# Flavones from *Combretum quadrangulare* Growing in Vietnam and Their Alpha-Glucosidase Inhibitory Activity

**DOI:** 10.3390/molecules26092531

**Published:** 2021-04-26

**Authors:** Thi-Bich-Ngoc Dao, Truong-Minh-Tri Nguyen, Van-Quy Nguyen, Thi-Minh-Dinh Tran, Nguyen-Minh-An Tran, Chuong Hoang Nguyen, Thi-Hoai-Thu Nguyen, Huu-Hung Nguyen, Jirapast Sichaem, Cong-Luan Tran, Thuc-Huy Duong

**Affiliations:** 1Department of Chemistry, University of Education, 280 An Duong Vuong Street, District 5, Ho Chi Minh City 72711, Vietnam; ngocdaosph@gmail.com (T.-B.-N.D.); nguyentruongminhtri99@gmail.com (T.-M.-T.N.); nguyenvanquysphoa@gmail.com (V.-Q.N.); 2Department of Biology, Ho Chi Minh City University of Education, 280 An Duong Vuong Street, District 5, Ho Chi Minh City 72711, Vietnam; dinhttm@hcmue.edu.vn; 3Industrial University of Ho Chi Minh City, Ho Chi Minh City 71420, Vietnam; trannguyenminhan@iuh.edu.vn; 4University of Science, Vietnam National University, Ho Chi Minh City 72711, Vietnam; nhchuong@hcmus.edu.vn; 5Faculty of Basic Sciences, University of Medicine and Pharmacy at Ho Chi Minh City, 217 Hong Bang Street, District 5, Ho Chi Minh City 72714, Vietnam; nguyenthihoaithu@ump.edu.vn; 6Faculty of Technology, Van Lang University, 45 Nguyen Khac Nhu, District 1, Ho Chi Minh City 71013, Vietnam; hung.nh@vlu.edu.vn; 7Research Unit in Natural Products Chemistry and Bioactivities, Faculty of Science and Technology, Thammasat University Lampang Campus, Lampang 52190, Thailand; jirapast@tu.ac.th; 8Faculty of Pharmacy and Nursery, Tay Do University, Can Tho 94000, Vietnam

**Keywords:** *Combretum quadrangulare* Kurz, flavonoid, alpha-glucosidase, antibacterial, molecular docking

## Abstract

*Combretum quadrangulare* Kurz is widely used in folk medicine in Eastern Asia and is associated with various ethnopharmacological properties including hepatoprotective, antipyretic, analgesic, antidysenteric, and anthelmintic activities. Previous phytochemical investigations reported the presence of numerous triterpenes (mostly cycloartanes, ursanes, lupanes, and oleananes) along with dozens of flavonoids. However, the extracts of *C. quadrangulare* and isolated flavonoids have not been evaluated for their alpha-glucosidase inhibition. In the frame of our efforts dedicated to the chemical investigation of Vietnamese medicinal plants and their biological activities, a phytochemical study of the MeOH extract of the leaves of *C. quadrangulare* using bioactive guided isolation was undertaken. In this paper, the isolation and structure elucidation of twelve known compounds, 5-hydroxy-3,7,4′-trimethoxyflavone (**1**), ayanin (**2**), kumatakenin (**3**), rhamnocitrin (**4**), ombuin (**5**), myricetin-3,7,3′,5′-tetramethyl ether (**6**), gardenin D (**7**), luteolin (**12**), apigenin (**13**), mearnsetin (**14**), isoorientin (**15**), and vitexin (**16**) were reported. Bromination was applied to compounds **2** and **3** to provide four new synthetic analogues **8**–**11**. All isolated and synthesized compounds were evaluated for alpha-glucosidase inhibition and antibacterial activity. Compounds **4** and **5** showed moderate antibacterial activity against methicillin-resistant *Staphylococcus aureus* while others were inactive. All compounds failed to reveal any activity toward extended spectrum beta-lactamase-producing *Escherichia coli*. Compounds **2**, **4**, **6**–**9**, and **11**–**14** showed good alpha-glucosidase inhibition with IC_50_ values in the range of 30.5–282.0 µM. The kinetic of enzyme inhibition showed that **8** and **11** were noncompetitive type inhibition against alpha-glucosidase. In silico molecular docking model indicated that compounds **8** and **11** were potential inhibitors against enzyme *α*-glucosidase.

## 1. Introduction

*Combretum quadrangulare* Kurz is widely used in folk medicine in Eastern Asia. The traditional uses have been corroborated by in vitro data of hepatoprotective, antipyretic, analgesic, antidysenteric, and anthelmintic properties [1,2]. In Vietnam, this plant was used as antihelmintic and antihepatitis agents [1,2]. Although former chemical studies on *C. quadrangulare* reported the presence of hundreds of triterpenes (cycloartanes, ursanes, lupanes, and oleananes), over 10 flavonoids were found from this bio-source [3,4,5,6,7,8]. Among the different organs of the plant, the leaves of *C. quadrangulare* attracted chemists to investigate. The crude MeOH extracts of the Vietnamese *C. quadrangulare* (leaves and seeds) showed a significant hepatoprotective effect, xanthine oxidase inhibition, and cytotoxicity against several cancer cell lines [6,7,9,10]. Up to now, ten flavones—combretol (from seeds and flowers); ayanin; 5-hydroxy-2-(4′-hydroxy-3′,5′-dimthoxyphenyl)-3,7-dimethoxy-4*H*-1-benzopyran-4-one (from flowers); kumatakenin; isokaempferide; 5,7,4′-trihydroxy-3,3′-dimethoxyflavone, 5,4′-dihydroxy-3,7,3′-trimethoxyflavone; isoorientin; isovitexin 4′-methyl ether (from leaves); and vitexin (from leaves and seeds)—were isolated from the Vietnamese native plant [2,6,7,9]. These compounds showed a strong inhibitory effect on TNF-α-induced cell death and mild to strong cytotoxicity against several cancer cell lines: 26-L5, HT-1080, HeLa, and A549 [2]. Nonetheless, the alpha-glucosidase inhibition and antibacterial activity of the extracts of *C. quadrangulare* and the isolated compounds have not been reported. As a continuation of our research focused on the diversity of bioactive metabolites from Vietnamese medicinal plants and their biological activity [10,11,12,13], a phytochemical study of the MeOH extract of the leaves of *C. quadrangulare* using bioactive guided isolation was undertaken. In this paper, the isolation and structural elucidation of twelve known flavonoids (**1**–**7**, **12**–**16**) are reported. Moreover, the electrophilic bromination was applied to compounds **2** and **3** to obtain four new analogues 8–11. Their structures were elucidated by spectroscopic data analysis and comparison with literature data. The isolated compounds were evaluated for alpha-glucosidase inhibition and molecular docking studies were performed to elucidate the mechanisms of inhibition. All compounds were further assayed for their in vitro antibacterial activity against methicillin-resistant *Staphylococcus aureus* and extended spectrum beta-lactamase-producing *Escherichia coli* (ESBL-producing *E. coli*).

## 2. Results and Discussion

### 2.1. Phytochemical Study and Derivatization of Compounds 2 and 3

The crude MeOH extract was successively partitioned into *n*-hexane, *n*-hexane:EtOAc (1:1, *v*/*v*), and EtOAc to provide extracts H, HEA, EA, and M**,** respectively. These fractions were evaluated for α-glucosidase inhibition. The most active extract was HEA. This was fractionated by column chromatography (CC), providing the fractions P1–P9. Of these, P2 and P9 showed the strongest biological activity (see Appendix A). Isolation and purification were performed on these fractions (see Experimental Section), affording compounds **1**–**7** and **12**–**16**.

The crude extract were fractioned and purified using chromatographic techniques to furnish 5-hydroxy-3,7,4′-trimethoxyflavone (**1**) [14], ayanin (**2**) [15], kumatakenin (**3**) [16], rhamnocitrin (**4**) [17], ombuin (**5**) [18], myricetin-3,7,3′,5′-tetramethyl ether (**6**) [19], gardenin D (**7**) [20], luteolin (**12**) [21], apigenin (**13**) [22], mearnsetin (**14**) [23], vitexin (**15**) [24], and isoorientin (**16**) [25] (Figure 1). These were identified by comparing their NMR spectra with published data (Appendix A). Among them, compounds **4**–**7** and **12**–**14** were reported for the first time from *C. quadrangulare.*

Compounds 2 and 3 were selected for bromination to obtain new flavones (Scheme 1). The reactions were conducted using hydrogen peroxide and potassium bromide in acetic acid (Scheme 1). As a result, compound 8 was prepared from 2 while products 9–11 were derived from 3. The isolated yields of these compounds were 35–93%. The 1H NMR spectrum of 8 displayed two meta-coupled protons at δ_H_ 7.88 and 8.18, three methoxy groups at δ_H_ 3.80 and 3.97 (x2), and one hydrogen-bond hydroxyl group at δ_H_ 15.71. NMR data of **8** and its mother compound **2** was highly similar. The differences are the disappearance of three aromatic protons (H-6, H-8, and H-5′), indicating that three positions were brominated. Altogether, the chemical structure of **8** was elucidated as shown. The identification of compounds **9**–**11** was readily established on the basis of their ^1^H, ^13^C NMR, and HRESI mass spectra (Appendix A).

### 2.2. Alpha-Glucosidase Inhibitory Activity of Isolated Compounds

The in vitro alpha-glucosidase inhibitory activity of **1**–**16** was evaluated (Table 1). Compounds **2**, **4**, **6**–**9**, and **11**–**14** exhibited good inhibition of alpha-glucosidase with the IC_50_ values were in the range of 30.5 to 282.0 μM, greater than the standard, acarbose (IC_50_ 332.5 μM). The C-3-substituted flavones **1**–**6** were relatively weaker than C-3-non-substituted analogues **7**, **12**, and **13**, indicating the important role of the 3-H substituent in biological activity among flavones. As regards to the synthesized products, compounds **8**–**11** were more active than their parent compounds **2** and **3**. The 6-brominated and 8-brominated products **9** and **10** were slightly stronger than their parent compound (**3**) whilst **11** significantly increased the activity (IC_50_ 30.5 μM). This indicated the activity preference for the 3′/5′-bromo positions. Similarly, 6,8,5′-tribromoayanin (**8**) was stronger than the starting material **2** (IC_50_ 87.1 μM). It is worth noting that the number of bromine atoms of the B-ring affected the biological activity. 

### 2.3. Inhibition Type and Inhibition Constants of the Compounds **8** and **11** on Alpha-Glucosidase 

In order to examine the inhibition mechanism of compound **11**, their activity was measured at the different concentration of *p*NPG. The Lineweaver–Burk plots of a kinetic study of **8** and **11** showed linearity at each concentration examined (0, 4.96, 9.92, and 19.84 µM for **8** and 0, 21.62, 43.03, and 86.06 µM for **11**), which all intersected the *x*-axis in the second quadrant (Figure 2). The kinetic analysis revealed that V_max_ decreased while Km remained constant, which showed that compounds **8** and **11** acted as noncompetitive inhibitors. The inhibition constants (*Ki*) of **8** was 39.82 µM and that of **11** was 198.87 µM.

### 2.4. Antibacterial Activity of Isolated Compounds

All compounds were evaluated for their in vitro antibacterial activity using the agar well diffusion method against methicillin-resistant *Staphylococcus aureus* (MRSA). Compounds **4** and **5** inhibited MRSA with diameters of inhibition zones of 14 and 15 mm, respectively, at the quantity of 50 μg for each compound, compared to that of the positive control, apramycin (24 mm). The others are inactive. All compounds failed to reveal any activity against ESBL-producing Escherichia coli.

### 2.5. In Silico Molecular Docking Model

The calculation results of molecular docking model of high active compound, **8**, **11**, and Acarbose based on autodock tools have been built in Figure 3, Figure 4, Figure 5, Figure 6, Figure 7, Figure 8 and Figure 9, Appendix A. In an in silico docking model, the most stable conformation of ligand **11** was bound to the active pocket on target enzyme 4J5T. Those interactions were assessed to be very strong because the values of Free Energy of Binding ΔG and the inhibition constant, *K_i_* between the most stable ligand **11** and target enzyme 4J5T have been calculated: −9.45 Kcal/mol and 0.118 µM, as shown in Figure 3 and Appendix A. As indicated in Figure 4 and Appendix A, four hydrogen bonds linked from residuals amino acids of target enzyme to functional groups of ligand such as A:ARG428:N-11:O (2.91 Å), A: ASN448:N-**11**: Br (3.14 Å), 11:H-A:ILE362:O (1.83 Å), and **11**:H-A: GLN447:O (2.10 Å). Those hydrogen bonds were formed with hydrogen atoms of phenolic hydroxyl and bromine atoms of aromatic rings. For the hydrogen linked from the bromine atom of ligand to the oxygen atom of ILE362 of amino acid of A chain, the receptor was the strongest hydrogen, 1.83 Å among them. Those hydrogen bonds had hydrophilic interactions (green areas in Figure 3). As shown in Figure 5, the significant interactions were formed from active sites on the receptor to the most stable ligand such as classical hydrogen bonds; for instance, Asn 448 to bromo of phenyl ring, Arg 428 to oxygen of OH group of phenolic hydroxy, Val 446 to bromo atom of phenyl ring, Ile 362 to hydrogen atom of OH group of phenyl ring, and Gly 447 to hydrogen atom of phenolic hydroxy. Another non classical hydrogen bond was from Lys 363 to oxygen of hydroxy of phenolic hydroxy. The halogen interactions linked from Glu 429 and Gln 442 to bromine atoms of phenyl rings, respectively. The hydrophobic ligand interactions formed pi-sima from alkenyl group to Leu 364, pi-alkyl from Leu 364 to pi system of aromatic ring, and alkyl from Lys 363, Arg 428, and Phe 444 to the bromine atoms of aromatic rings. Other residual interactions were determined from Glu 361, Glu 443, and Gln 445, which linked weak interactions to the wall of cells. Briefly, the ligand 11 was the potential delivery drug because it identified well functional groups (via hydrogen bonds, halogen bonds), cap groups (via pi-sigma, alkyl, pi-alkyl interactions), and other Van Der Waals interactions, which bound weak to the wall of cells, as shown in Figure 5. Other the secondary interactions, which formed between ligand 11 and receptor 4J5T, were hydrogen bond, steric, and overlap interactions. Those interactions have established the stable interactions of the conformation ligand and receptor, as indicated in Figure 6. Those steric interactions (light blue), which linked from residual amino acids of A chain of 4J5T to active site atoms on 11 were Ile 362, Val 446, Glu 443, Phe 444, Arg 428, and Asn 448. The hydrogen bond (brown color) formed from Ile 362 and Gln 447 to the most stable conformation ligand **11**. The steric interactions show violet circles on atoms of the conformation ligand. The bigger the effect of the steric, the bigger the sizes of the violet circles were, as shown in Figure 6. The interaction profile between the most stable ligand 8 and target enzyme is built in Figure 7, Figure 8 and Figure 9, Appendix A. As shown in Figure 7, the most stable conformation ligand 8 was selected to link to the target enzyme on enzyme pocket. Those interactions were assessed with the value of Free Energy of Binding ΔG and the inhibition constant K_i_, as shown in Figure 7, and calculated as −8.83 Kcal/mol and 0.337 µM. As indicated in Figure 8, two classical hydrogen bonds, which formed from hydrogen phenolic hydroxy to residual amino acid of target enzyme 4J5T, were established for instant 8:H-A:ILE362:O (2.05 Å) and **8**:H-A:PHE444:O (1.98 Å). They were the significant hydrophilic interactions, forming between ligand **8** and target enzyme. The interaction profile of the most stable conformation ligand **8** and enzyme is presented on a 2D diagram shown in Figure 9. As shown in Figure 9, ligand **8** has potential for drug delivery because it can be identified in a cap group (via ligand interactions such as pi-sigma: from Leu 364 to pi bound of alkenyl conjugation with ketone, mixed alkyl or pi-alkyl: from Leu: 364 to aromatic ring of ligand), linker (from Gln 447, Arg 428, Ile 362, Ile 364, Ile 451, Lys 363, Leu 365, and Val 446 to methoxy groups, bromine atoms of ligand), and functional groups (Phe 444, and Ile 362). Other residual interactions, Van Der Waal interactions, formed around ligands (linked wall cell) such as Gln 442, Glu 443, Glu 429, Asn 448, and Glu 361). Both candidates have potential for drug delivery but **11** is superior to **8** because of the values of ΔG and Ki.

## 3. Materials and Methods

### 3.1. Source of the Plant Material

Leaves of *Combretum quadrangulare* were collected in Duc Hoa, Long An Province in March and April 2020. The plant was identified as *Combretum quandrangulare* Kurz by Dr. Tran Cong Luan, Tay Do University, Vietnam. A voucher specimen (No UE-002) was deposited in the herbarium of the Department of Organic Chemistry, Faculty of Chemistry, Ho Chi Minh University of Education, Ho Chi Minh City, Vietnam.

### 3.2. Isolation and Structure Elucidation of the Compounds

Gravity column chromatography was performed on silica gel 60 (0.040–0.063 mm, Merck, Darmstadt, Germany). TLC for checking chromatographic patterns of fractions and isolated compounds was carried out on silica gel 60 F_254_ (Merck) and spots were visualized by spraying with 10% H_2_SO_4_ solution followed by heating. Specific rotations were obtained on a Jasco P-1010 polarimeter (Oklahoma City, OK, USA). The HR-ESI-MS were recorded on a MicroOTOF-Q mass spectrometer (Bruker, Billerica, MA, USA). The NMR spectra were measured on a Bruker Avance 500 MHz spectrometer (Bruker, Billerica, MA, USA).

### 3.3. Isolation

Dried leaves of *C. quadrangulare* (11 kg) were crushed and extracted with MeOH (3 × 30 L) at an ambient temperature for 24 h. The filtrated solution was evaporated to dryness under reduced pressure to obtain a crude extract (118.4 g). This crude was successively partitioned by *n*-hexane, *n*-hexane: EtOAc (1:1, *v*/*v*), EtOAc, to afford **H** (29.1 g), **HEA** (160.3 g), **EA** (30.0 g), and **MeOH** (**M**, 12.0 g), respectively. Fraction **HEA** (160.3 g) was subjected to silica gel column chromatography (CC), using an isocratic mobile phase consisting of *n*-hexane: EtOAc: acetone (5:1:1, *v*/*v*/*v*) to obtain nine fractions: **P1** (4.95 g), **P2** (9.72 g), **P3** (6.94 g), **P4** (4.82 g), **P5** (5.69 g), **P6** (4.23 g), **P7** (3.2 g), **P8** (4.15 g), and **P9** (3.9 g). Fraction **P2** (9.72 g) was subjected to silica gel column chromatography, using the solvent system of *n*-hexane: EtOAc: acetone (5:1:1, *v*/*v*/*v*) to obtain fractions **T1** (1.8 g), **T2** (600.0 mg), **T3** (900.0 mg), **T4** (2.0 g), **T5** (1.1 g), and **T6** (1.3 g). 

Fraction **T1** (1.8 g) was submitted to silica gel CC, eluted by the solvent system *n*-hexane: CHCl_3_: EtOAc: acetone (3:2:1:1, *v*/*v*/*v*/*v*) to obtain four fractions: **T1.1** (310.0 mg), **T1.2** (560.0 mg), **T1.3** (230.0 mg), and **T1.4** (190.0 mg). Fraction **T1.2** was rechromatographed, eluted with the same solvent system to obtain three compounds: **2** (55.2 mg), **7** (5.1 mg), and **14** (2.1 mg). Fraction **T5** (1.1 g) was submitted to silica gel CC using the mixture of *n*-hexane: CHCl_3_: EtOAc: acetone (3:2:2:2, *v*/*v*/*v*/*v*) as a mobile phase to obtain three fractions, **T5.1** (210.0 mg), **T5.2** (350.0 mg), and **T5.3** (150.0 mg). Fraction **T5.1** was rechromatographed and eluted with the same solvent system to obtain two compounds: **1** (7.4 mg) and **3** (45.0 mg). Fraction **T5.2** was subjected to silica gel CC, eluted with the solvent system *n*-hexane: CHCl_3_: EtOAc: acetone (4:3:3:1, *v*/*v*/*v*/*v*) to afford three compounds: **6** (4.7 mg), **12** (3.8 mg) and **13** (6.8 mg). Fraction **T5.3** was rechromatographed using an isocratic mobile phase consisting of a *n*-hexane: CHCl_3_: EtOAc: acetone: H_2_O (2:2:1:1:0.01, *v*/*v*/*v*/*v*/*v*) to gain two compounds, **4** (5.1 mg) and **5** (4.7 mg). Fraction **P9** was applied to Sephadex LH-20 gel chromatography and eluted with methanol to afford three fractions, **P9.1**–**P9.3**. Fraction **P9.3** (601 mg) was fractionated by silica gel CC to afford compounds **15** (1.9 mg) and **16** (2.7 mg).

### 3.4. General Procedure to Synthesize Compounds **8**–**11**

In 4 mL of acetic acid, ayanin (**2**, 10 mg, 0.029 mmol) and sodium bromide (14.9 mg, 0.145 mmol) were dissolved at room temperature under stirring. The reaction was added to 0.02 mL (0.196 mmol) of 30% hydrogen peroxide. The reaction was conducted for 30 min and was periodically monitored by TLC. After neutralizing it with saturated sodium hydrogen carbonate, the mixture was extracted with ethyl acetate-water (1:1, *v*/*v*) to gain an organic layer. This was pooled, washed with brine, and dried over anhydrous Na_2_SO_4_. The residue was further absorbed onto column chromatography successively using the gradient system of *n*-hexane: chloroform (1:4, *v*/*v*) to afford **8** (14.9 mg). The procedure followed the previous report with modifications [26].

In 4 mL of acetic acid, kamatakenin (**3**, 10 mg, 0.032 mmol) and sodium bromide (16.4 mg, 0.160 mmol) were dissolved at room temperature under stirring. The reaction was added to 0.02 mL (0.196 mmol) of 30% hydrogen peroxide and was conducted in 30 min. The work-up followed the same procedure as mentioned previously to obtain the residue. The residue was further absorbed onto column chromatography successively using the gradient system of *n*-hexane: chloroform (1:4, *v*/*v*) to afford **11** (18.6 mg). The ratio of **3** and sodium bromide was modified to 1:1 and the reaction was repeated following the previously mention procedure. The reaction was conducted in 1 h and compounds **9** (4.4 mg) and **10** (6.0 mg) were obtained after silica gel CC. 

#### 3.4.1. 6,8,5′-Tribromoayanin (**8**)

Isolated yield 89.0%; light yellow powder. UV (MeOH) λ_max_ (log*ε*) 217 (4.59), 257 (4.42), 360 (4.34) nm; IR cm^−1^ (KBr): 3389, 1670, 1626, 1575, 1405, 1372**.**
^1^H NMR (500 MHZ, Acetone–*d_6_) δ*_H_ 15.72 (*s*, 1H, 5-OH), 8.18 (*d*, 1H, J = 2.0 Hz, H-6′), 7.87 (*d*, 1H, J = 1.5 Hz, H-2′), 3.99 (*s*, 3H, 7-OCH_3_), 3.97 (*s*, 3H, 4′-OCH_3_), 3.79 (*s*, 3H, 3-OCH_3_). ^13^C NMR (125 MHz, Acetone–*d_6_*) *δ*_C_ 178.8 (C-4), 172.6 (C-7), 161.4 (C-5), 160.6 (C-9), 156.6 (C-2), 152.3(C-4′), 149.5 (C-3′), 139.7 (C-3), 127.3 (C-6), 122.1 (C-1′), 110.9 (C-2′), 109.8 (C-10), 97.0 (C-5′), 94.0 (C-6), 88.7 (C-8), 61.6 (7-CH_3_), 60.2 (3-CH_3_), 56.6 (4′-CH_3_). HR-ESI-MS *m*/*z* 576.8133 (calcd. for C_18_H_12_O_7_Br_3_, 576.8133).

#### 3.4.2. 8-Bromokamatakenin (**9**)

Isolated yield 35.1%; light yellow powder. UV (MeOH) λ_max_ (log*ε*) 206 (4.32), 258 (4.29), 360 (4.16) nm; IR cm^−1^ (KBr): 3389, 1670, 1626, 1575, 1405, 1372. ^1^H NMR (DMSO–*d_6_) δ*_H_ 12.85 (*s*, 1H, 5-OH), 10.39 (s, 1H, 4′-OH), 8.07 (*d*, 2H, H-2′,6′), 6.98 (*d*, J = 9.0Hz, H-3′,5′), 3.97 (*s*, 3H, 7-OCH_3_), 3.82 (*s*, 3H, 3-OCH_3_). ^13^C NMR (125 MHz, DMSO– *d_6_*) *δ*_C_ 178.0 (C-4), 161.3 (C-7), 160.7 (C-5), 160.6 (C-4′), 155.8 (C-2), 151.7 (C-9), 137.6 (C-3), 130.3 (C-2′,6′), 120.4 (C-1′), 115.7 (C-3′,5′), 105.5 (C-10), 96.1 (C-6), 87.0 (C-8), 59.7 (3-OCH_3_), 57.3 (7-OCH_3_). HR-ESI-MS *m*/*z* 390.9831 (calcd. for C_17_H_12_O_6_Br, 390.9817).

#### 3.4.3. 6-Bromokamatakenin (**10**)

Isolated yield 47.6%; light yellow powder. UV (MeOH) λ_max_ (log*ε*) 208 (4.19), 271 (4.02), 343 (3.99) nm; IR cm^−1^ (KBr): 3389, 1670, 1626, 1575, 1405, 1372. ^1^H NMR (DMSO*–d_6_) δ*_H_ 13.47 (*s*, 1H, 5-OH), 10.34 (s, 1H, 4′-OH), 8.02 (*d*, 2H, H-2′,6′), 7.03 (*s*, H-8), 6.96 (*d*, J = 9.0Hz, H-3′,5′), 3.98 (*s*, 3H, 7-OCH_3_), 3.82 (*s*, 3H, 3-OCH_3_). ^13^C NMR (125 MHz, DMSO*– d_6_*) *δ*_C_ 177.6 (C-4), 161.5 (C-5), 160.8 (C-7), 160.5 (C-4′), 156.3 (C-9), 155.3 (C-2), 138.9 (C-3), 130.3 (C-2′,6′), 120.3 (C-1′), 115.7 (C-3′,5′), 105.6 (C-10), 99.7 (C-6), 95.0 (C-8), 59.7 (3-OCH_3_), 57.3 (7-OCH_3_). HR-ESI-MS *m*/*z* 390.9843 (calcd. for C_17_H_12_O_6_Br, 390.9817)

#### 3.4.4. 6,8,3′,5′-Tetrabromokamatakenin (**11**)

Isolated yield 93.0%; light yellow powder. UV (MeOH) λ_max_ (log*ε*) 211 (4.46), 254 (4.11), 416 (4.18) nm; IR cm^−1^ (KBr): 3389, 1670, 1626, 1575, 1405, 1372. ^1^H NMR (DMSO–*d_6_) δ*_H_ 13.41 (*s*, 1H, 5-OH), 8.28 (*s*, 2H, H-2′,6′), 3.92 (*s*, 3H, 7-O CH_3_), 3.87 (*s*, 3H, 3-OCH_3_).^13^C NMR (126 MHz, DMSO*–d_6_*) *δ*_C_ 177.7 (C-4), 166.2 (C-7), 161.7 (C-5), 161.2 (C-4′), 159.2 (C-9), 156.9 (C-2), 138.5 (C-3), 132.2 (C-2′, 6′), 121.2 (C-1′), 112.2 (C-3′,5′), 108.7 (C-10), 99.3 (C-6), 94.7 (C-8), 60.7 (3-O CH_3_), 59.5 (7-O CH_3_). HR-ESI-MS *m*/*z* 624.7127 (calcd. for C_17_H_9_O_6_Br_4_, 624.7133).

### 3.5. Alpha-Glucosidase Inhibition Assay

The alpha-glucosidase (0.2 U/mL) and substrate (5.0 mM *p*-nitrophenyl-α-D-glucopyranoside) were dissolveed in 100 mM of pH 6.9 sodium phosphate buffer [27]. The inhibitor (50 µL) was preincubated with alpha-glucosidase at 37 °C for 20 min, and then the substrate (40 µL) was added to the reaction mixture. The enzymatic reaction was carried out at 37 °C for 20 min and stopped by adding 0.2 M of Na_2_CO_3_ (130 μL). Enzymatic activity was quantified by measuring absorbance at 405 nm. All samples were analyzed in triplicate at five different concentrations around the IC_50_ values, and the mean values were retained. The inhibition percentage (%) was calculated by the following equation:Inhibition (%)=[1−(Asample/Acontrol)]×100..

### 3.6. Inhibitory Type Assay of **8** and **11** on Alpha-Glucosidase

The mechanisms of inhibition of alpha-glucosidase by **8** and **11** were determined by Lineweaver–Burk plots (Microsoft Excel 2010, Washington, WA, USA), using methods similar to those reported in the literature [28]. Enzyme inhibition due to various concentrations of the **8** and **11** compounds were evaluated by monitoring the effects of different concentrations of the substrate. For Lineweaver–Burk double reciprocal plots 1/enzyme velocity (1/V) vs. 1/substrate concentration (1/[S]), the inhibition type was determined using various concentrations of *p*NPG (1 mM, 2 mM, and 4 mM) as a substrate in the presence of different concentrations of the test compounds (0; 4.96; 9.92 and 19.84 µM for **8** and 0; 21.62; 43.03 and 86.06 µM for **11**). The experiments were carried out in 3 replicates. The mixtures were incubated at 37 °C and the optical density was measured at 405 nm every 1 min for 30 min with the Clariostar Labtech microplate reader (Ortenberg, Germany). Optimal concentrations of the tested compound were chosen based on the IC_50_ value. The inhibition constants were obtained graphically from secondary plots (Microsoft Excel 2010, Washington, WA, USA).

### 3.7. Antibacterial Activity Assay

Methicillin-resistant *Staphylococcus aureus* (MRSA), extended spectrum beta-lactamase-producing Escherichia coli (ESBL-producing *E. coli*) and the agar well diffusion method were used to evaluate the antibacterial activity of the isolated compounds [29]. The strains were cultured in nutrient broth at 37 °C for 18 h and diluted with steriled 0.9% NaCl to obtain the 1.5 × 10^8^ CFU/mL bacterial solution. Then, 100 μL of the bacterial solution was spread on Mueller–Hinton agar (MHA) plate on which 8-mm wells were created aseptically with tips. The isolated compounds were dissolved in dimethyl sulfoxide at the concentration of 1 mg/mL and 50 μL of each compound solution was applied in each well. The plates were incubated at 37 °C for 16–18 h and the diameters of the inhibtion zones were measured. Dimethyl sulfoxide and apramycin were used as controls in this experiment.

### 3.8. Molecular Docking Study Method 

The calculations of the molecular docking model were performed according to the procedure in the supporting information file and followed the previous article [30]. 

## 4. Conclusions

From the Vietnamese plant *C. quadrangulare*, twelve alpha-glucosidase inhibitors were isolated and elucidated, including 5-hydroxy-3,7,4′-trimethoxyflavone (**1**), ayanin (**2**), kamatakenin (**3**), rhamnocitrin (**4**), ombuin (**5**), myricetin-3,7,3′,5′-tetramethyl ether (**6**), gardenin D (**7**), luteolin (**12**), apigenin (**13**), mearnsetin (**14**), isoorientin (**15**), and vitexin (**16**). To the best of our knowledge, compounds **4**–**7** and **12**–**14** were reported for the first time from *C. quadrangulare.* New synthetic analogues **8**–**11** were prepared via electrophic bromination of compounds **2** and **3**. Compounds **4** and **5** showed moderate antibacterial activity against methicillin-resistant *Staphylococcus aureus*. Compounds **2**, **4**, **6**–**9**, and **11**–**14** displayed good alpha-glucosidase inhibition, with IC_50_ values in the range of 30.5–282.0 µM. Among them, synthetic compounds revealed the strongest inhibition. The kinetic of enzyme inhibition showed that **8** and **11** were noncompetitive type inhibition against alpha-glucosidase. An in silico molecular docking model indicated that compounds **8** and **11** were potent alpha-glucosidase inhibitors. 

## Data Availability

The data presented in this study are available in Appendix A.

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
