# Peer review of "Flavones from Combretum quadrangulare Growing in Vietnam and Their Alpha-Glucosidase Inhibitory Activity"

_molecules, 2021, doi:10.3390/molecules26092531_

Round 1

Reviewer 1 Report

Interesting article on the isolation of new flavons from C. quadrangulare, the synthesis of some derivatives and their selected biological properties.
The English language is flawless, as is the work at a high level in terms of editing (one mistake - line 26 - unnecessary space in the word "phytochemical") and aesthetically. A big plus for presenting a compound isolation graph.

I miss explanation why the alpha-glucosidase inhibition activity and activity against S. aureus (MRSA) were determined for those compounds. Are there any literary premises? What about activity against E.coli? Please attach relevant explanations to the article.

Additionally, I do not know why the new bromo derivatives were synthesized. Why exactly such? Please also explain and attach them to the article.

For new compounds, please include UV-Vis spectra (in article and in Supplementary) and melting points.

Author Response

FLAVONES FROM COMBRETUM QUADRANGULARE GROWING IN VIETNAM AND THEIR ACTIVITIVES

We thank for the corrections of the Reviewers and have tried to adjust the manuscript.

All errors corrected by the Reviewer were adjusted and highlighted in yellow color in the manuscript.

Reviewer #1:

1, Interesting article on the isolation of new flavons from C. quadrangulare, the synthesis of some derivatives and their selected biological properties. The English language is flawless, as is the work at a high level in terms of editing (one mistake - line 26 - unnecessary space in the word "phytochemical") and aesthetically. A big plus for presenting a compound isolation graph.

The authors greatly appreciated the Reviewer for this positive comment.

  1. I miss explanation why the alpha-glucosidase inhibition activity and activity against S. aureus(MRSA) were determined for those compounds. Are there any literary premises? What about activity against E.coli? Please attach relevant explanations to the article.

Thank you very much for the valuable comment. All compounds were isolated from the most active fraction inhibiting alpha-glucosidase (bio-active guided isolation). The authors provided the IC50 values of all fractions in Table S6. For antibacterial, all compounds were tested because the experiments are available in our lab. All compounds were evaluated the activity against ESBL- E.coli but they are inactive. We inserted the results in the manuscript.

  1. Additionally, I do not know why the new bromo derivatives were synthesized. Why exactly such? Please also explain and attach them to the article.

Thank you very much for this remark. The bromination were chosen because in our experience, the yield of such reactions are very high. Thus, it is very suitable for the low amounts of the starting materials 2 and 3 (45-55 mg).

  1. For new compounds, please include UV-Vis spectra (in article and in Supplementary) and melting points.

The authors are grateful for the comment. We provided UV data in the manuscript. The authors could not record melting points of new compounds because all new compounds are amorphous powders, not crystals.

Reviewer 2 Report

The manuscript by Dao et al. reports the phytochemical study of the methanol extract of Combretum quadrangulare leaves, a species used in folk medicine in Eastern Asia. Twelve known flavones were isolated, some of them reported in this species for the first time. Two flavones were derivatized by bromination reaction to obtain four new derivatives. Spectroscopic data of the new derivatives were presented. All compounds were evaluated for alpha-glucosidase inhibition and antibacterial activity.In silico molecular docking studies were also performed.

The manuscript falls in the scope of Molecules, however, some revisions are necessary because there are several points that have to be changed, in order to improve the manuscript. Furthermore, the Authors should check the English language and correct some typos and grammar errors.

Title – The title should be revised to be more specific, as the only reported activity (the most important apart from the antibacterial) is the alpha-glucosidase inhibition.

Lines 24 and 25 – do not repeat the word “properties”

Line 28 (and line 52) – Reference 3 reported the isolation and synthesis of combretastatins from Combretum caffrum. This reference should be removed as it doesn’t refer to the study of C. quadrangulare.

The authors stated the use of bio-guided isolation to obtain the flavones (see line 76 as well). However. This is not described in the materials and methods. What kind of activities have you screened? Was it the alpha-glucosidase inhibition activity? What were the results obtained for the different extracts and fractions?

Line 52 – This sentence should be re-written.

Line 54 – What parts of the plant have been studied for these activities? The leaves?

Line 63 – Authors indicated that the alpha-glucosidase inhibition and antibacterial activity of the extracts of C. quadrangulare and isolated compounds have not been reported yet. On the other hand, they stated bio-guided isolation of the flavones, but these data are missing in the manuscript. This must be corrected throughout the manuscript for clarity reasons.

Line 76 – This subtitle should be corrected to: “Phytochemical study and derivatization of compounds 2 and 3

Line 88 – I don’t think that the isolation of 40 mg of compound could be considered a “large amount”. This sentence must be corrected. Furthermore, you haven’t performed a semi-synthesis. Compounds 2 and 3 were derivatized by electrophilic bromination reaction to yield new compounds or analogs.

Lines 89 – 92: this paragraph must be improved because all derivatives were obtained using the same reaction.

Lines 107 – 109: Do you mean tri-substituted?

Line105: What is the cut-off value for a compound to be considered as a “potent” inhibitor of alpha-glucosidase?

Line 110 – parent compound instead of “mother” compound

Table 1. Data from Standard deviation (SD) is missing

Lines 127 – 130: What was the value of the inhibition zone found for the positive control (apramycin) used?

Lines 259, 268, and 270 – the quantities of the obtained derivatives are missing.

Lines 256 – 257: this sentence must be revised.

Line 314: A reference is missing.

Lines 338 – 345 – section 3.7 should be located before the description of the isolation procedures (before 3.2)

The conclusion section needs an English revision for correction of grammar and typos.

Author Response

FLAVONES FROM COMBRETUM QUADRANGULARE GROWING IN VIETNAM AND THEIR ACTIVITIVES

We thank for the corrections of the Reviewers and have tried to adjust the manuscript.

All errors corrected by the Reviewer were adjusted and highlighted in yellow color in the manuscript.

  1.  

The authors are grateful for the comment. We provided UV data in the manuscript. The authors could not record melting points of new compounds because all new compounds are amorphous powders, not crystals.

Reviewer #2:

  1. The manuscript by Dao et al. reports the phytochemical study of the methanol extract of Combretum quadrangulare leaves, a species used in folk medicine in Eastern Asia. Twelve known flavones were isolated, some of them reported in this species for the first time. Two flavones were derivatized by bromination reaction to obtain four new derivatives. Spectroscopic data of the new derivatives were presented. All compounds were evaluated for alpha-glucosidase inhibition and antibacterial activity. In silico molecular docking studies were also performed. The manuscript falls in the scope of Molecules, however, some revisions are necessary because there are several points that have to be changed, in order to improve the manuscript. Furthermore, the Authors should check the English language and correct some typos and grammar errors.

The authors are indebted the Reviewer for this positive comment.

  1. Title – The title should be revised to be more specific, as the only reported activity (the most important apart from the antibacterial) is the alpha-glucosidase inhibition.

The authors are grateful for the comment. We changed the Title following the suggestion. The new title is: “FLAVONES FROM COMBRETUM QUADRANGULARE GROWING IN VIETNAM AND THEIR ALPHA-GLUCOSIDASE INHIBITORY ACTIVITIVES”

  1. Lines 24 and 25: do not repeat the word “properties”

It was done.

  1. Line 28 (and line 52): Reference 3 reported the isolation and synthesis of combretastatins from Combretum caffrum. This reference should be removed as it doesn’t refer to the study of C. quadrangulare.

The authors are grateful for the comment. It was removed.

  1. The authors stated the use of bio-guided isolation to obtain the flavones (see line 76 as well). However. This is not described in the materials and methods. What kind of activities have you screened? Was it the alpha-glucosidase inhibition activity? What were the results obtained for the different extracts and fractions?

Line 63 – Authors indicated that the alpha-glucosidase inhibition and antibacterial activity of the extracts of C. quadrangulare and isolated compounds have not been reported yet. On the other hand, they stated bio-guided isolation of the flavones, but these data are missing in the manuscript. This must be corrected throughout the manuscript for clarity reasons.

The authors are indebted for this comment. The main activity for screening is alpha-glucosidase inhibition. The authors provided the IC50 values of all fractions in Table S6. The authors also added the paragraph regarding the bio-active guided isolation.

  1. Line 52 – This sentence should be re-written.

It was done.

  1. Line 54 – What parts of the plant have been studied for these activities? The leaves?

They are leaves and seeds. We inserted the missing information in the sentence.

  1. Line 76 – This subtitle should be corrected to: “Phytochemical study and derivatization of compounds 2 and 3

Thank you very much for the suggestion. It was done.

  1. Line 88 – I don’t think that the isolation of 40 mg of compound could be considered a “large amount”. This sentence must be corrected. Furthermore, you haven’t performed a semi-synthesis. Compounds 2 and 3 were derivatized by electrophilic bromination reaction to yield new compounds or analogs.

Thank very much for the remark. We revised accordingly.

  1. Lines 89 – 92: this paragraph must be improved because all derivatives were obtained using the same reaction.

We revised accordingly. Thank you.

  1. Lines 107 – 109: Do you mean tri-substituted?

The authors apology for this mistake. We revised these words.

  1. Line105: What is the cut-off value for a compound to be considered as a “potent” inhibitor of alpha-glucosidase?

The authors apology for this wrong statement. All compounds are stronger than the standard but they are not potent. We changed the word.

  1. Line 110: parent compound instead of “mother” compound

It was replaced.

  1. Table 1. Data from Standard deviation (SD) is missing.

They were added.

  1. Lines 127 – 130: What was the value of the inhibition zone found for the positive control (apramycin) used?

The lacking information of the standard apramycin were provided in the paragraph. Thank you very much.

  1. Lines 259, 268, and 270 – the quantities of the obtained derivatives are missing.

They were done.

  1. Lines 256 – 257: this sentence must be revised.

It was done.

  1. Line 314: A reference is missing.

The missing references were added.

  1. Lines 338- 345 section 3.7 should be located before the description of the isolation procedures (before 3.2)

Thank you very much for the suggestion. It was done.

  1. The conclusion section needs an English revision for correction of grammar and typos.

The authors greatly appreciated this comment. We revised following the suggestion.
